# Impact of a Starch Hydrolysate on the Production of Exopolysaccharides in a Fermented Plant-Based Dessert Formulation

**DOI:** 10.3390/foods12203868

**Published:** 2023-10-22

**Authors:** Aldjia Ait Chekdid, Cyril J. F. Kahn, Béatrice Lemois, Michel Linder

**Affiliations:** 1Université de Lorraine, LIBio, F-54000 Nancy, France; a.aitchekdid@gmail.com (A.A.C.); cyril.kahn@univ-lorraine.fr (C.J.F.K.); 2St-Hubert SA, 13-15 Rue du Pont des Halles, F-94150 Rungis, France; beatrice.lemois@sthubert.fr

**Keywords:** fermentation, starch, hydrolysis, exopolysaccharide, flour, texture

## Abstract

Plant-based desserts are becoming increasingly popular with and appreciated by consumers. However, they are limited by the choice of ingredients, which are often expensive and unstable with a random texture. Therefore, the aim of the research is to propose a new product that offers an advantageous texture and flavour in a fermented dessert based on a flour mix supplemented with an enzymatic hydrolysate. This study involved the development of two processes: (i) an enzymatic hydrolysis of oat flour and (ii) a fermentation of a flour mixture (oat, chickpea, and coconut) by lactic acid bacteria (*Lactobacillus delbrueckii* subsp. *bulgaricus* and *Streptococcus thermophilus*). The result of the oat flour hydrolysate shows a significant decrease in starch after 60 min of reaction, followed by an increase in sugar content. During 23 days of storage at 4 °C, the formulations used showed post-acidification, water retention capacity decrease, and hardness increase related to the hydrolysate rate (*p* < 0.05). All formulations allowed the viability of lactic bacteria (over 5 log_10_ CFU/mL) and verified their ability to produce exopolysaccharides (0.23–0.73 g/100 g). The prototyping of such a product represents a key step in meeting the growing demand for plant-based alternatives, with qualitative sensory characteristics without additives.

## 1. Introduction

Several food industries are contributing to a positive food transition. Their objective is to bring quality food to consumers while reducing their environmental impact. Plant-based desserts are one of the products that are currently gaining traction in the market. These products already have widespread usage and popularity in Asia and Africa. In Europe, the demand for these plant-based foods is also on the rise due to the concerns of consumers about the impact of their dietary choices on climate change and health [1,2].

Cereal-based products developed with precise lactic acid bacteria (LAB) strains are responding to this growing demand for healthier and more diverse food options. Furthermore, cereals constitute a group of seed crops grown worldwide, forming the backbone of the human diet. They are significant sources of carbohydrates, vitamins, minerals, and fibre, despite having low protein and fat content in comparison to milk and legumes [3,4]. Due to their composition, oats are one of the most suitable ingredients for desserts based on plant products. The starch content of oats can range from 60% to 65%, while the protein content varies from 9% to 15%. Oats also contain β-glucan, a dietary fibre that is between 2% and 8%. Oat milk as a soluble extract of oat flakes, is already consumed daily as a lactose- and gluten-free product due to its high content of dietary fibres known for their beneficial effects on health [5,6]. Therefore, combining cereals with leguminous crops has frequently been advised to produce low-cost, protein-rich foods and improve the balance of macronutrients and micronutrients [7]. Chickpeas can be used as an alternative in fermented plant-based beverages. They are one of the oldest cultivated legumes with an important source of high-quality protein that can reach 22%. Studies have examined the impact of various microorganisms on the fermentation of chickpeas, aiming to enhance microbial growth and improve the texture. The chickpea desserts were characterised by a fermentability comparable to soy milk, and chickpeas improved nutritional and sensory qualities [8,9,10]. In addition, coconut flour as a byproduct, contains elevated levels of fibre and protein and high concentrations of fermentable simple carbohydrates that can be used in food. It is the resulting product of coconut meat produced during the extraction of coconut milk or oil. It is dried and degreased to obtain a fine powder. Coconut is considered a nutraceutical food because of its many health benefits [11,12,13].

Starch, which is a complex of D-glucose units, is one of the most abundant carbohydrates in the biosphere and the main reserve of the storage organs of cereals, roots, tubers, and legumes. It consists of two glucose polymers, amylose and amylopectin, representing about 98–99% of the granule’s dry weight. The main function of starch in many products is to modify texture that ultimately contributes to the quality of processed foods [14]. Starch is used to uphold the quality and moisture and to regulate the mobility of the stored food products. Consequently, the functionality of starch depends on the physical organisation of these two main macromolecules within the granular structure, their ratio and size distribution, which vary depending on the botanical source of the starch [9,15,16]. The food industry commonly employs chemical, physical, and/or biochemical modifications of starches to fulfil their structural and functional demands. These modifications alter or fragment all or part of the molecules of starch, increasing or decreasing properties like viscosity, solubility, swelling, gelation, and digestibility. [17,18]. Enzymatic hydrolysis is widely used to modify starch properties. Starch hydrolysates are often used as substrates in many glucose-based fermentation processes and in a range of food and beverage industries [19]. Enzymes that hydrolyse starch (amylase or amylolytic enzymes) have received a lot of attention due to biotechnological applications and economic advantages. In industry, most processes use free enzymes, such as α-amylases, β-amylases, and glycoamylases, which are inactivated at the end of the reaction [20]. When starch is converted, enzymatic hydrolysis typically takes place in three stages. During the first stage, known as gelatinisation, the intermolecular bonds of starch are broken apart by heat in the presence of water. This process allows for the breakdown or rupture of starch granules, making them vulnerable to enzyme attack. During the second stage, known as liquefaction, starch is transformed into oligosaccharides, polysaccharides, or maltodextrin by the action of α-amylase, which also leads to a decrease in viscosity. In the third stage, saccharification occurs by glucoamylase or pullulanase, producing a combination of glucose and maltose [21].

The fermentation of plant-based desserts by LAB can be recognised as a suitable tool for improving flavour and texture formation through secondary reactions [22]. Lactic acid bacteria synthesise during fermentation molecules called exopolysaccharides. Exopolysaccharides (EPSs) are extracellular polysaccharides of high molecular weight between 10^3^ and 2.5 × 10^6^ Da. Generally, they are composed of monosaccharides and noncarbohydrate substitutes, such as acetate, phosphate, pyruvate, and succinate. EPSs form through the polymerisation of sugar subunits and can consist of either repeated glucose or fructose subunits (homopolysaccharides) or two or more different subunits (heteropolysaccharides). Therefore, the production of EPSs mostly depends on the sugar content in the medium, although it is influenced by the presence of micronutrients (e.g., minerals acting as enzyme cofactors) and the environmental conditions (temperature and incubation time) [22,23,24]. The major importance of EPSs is their power to improve texture by increasing the viscosity of the liquid substrate. The effects of EPSs also include improving mouthfeel, limiting syneresis, and increasing gel firmness, based on its ability to bind water and interact with proteins [4,24]. Mäkinen et al. [25] stated that *Streptococcus thermophilus* and *Lactobacillus delbrueckii* subsp. *bulgaricus* are strains that produce EPSs, which allow the viscosity to become comparable to dairy yoghurt when glucose is used as a carbon source in a fermented plant-based product [26].

The aim of this study was to develop a novel fermented food using chickpea, oat, and coconut flours and to investigate the effect of partially hydrolysed oat starch on the physicochemical properties and exopolysaccharide formation.

## 2. Materials and Methods

### 2.1. Starch Hydrolysis

#### 2.1.1. Enzyme Selection

Starch hydrolysis was made using a mixture of two enzymes from the Novozymes^®^ brand (Univar Solutions, Montreuil, France):

Enzyme 1: BAN^®^ 480L is a bacterial alpha-amylase that hydrolyses the (1,4) alpha-D-glycosidic bonds of starch polysaccharides and reduces the viscosity during starch gelatinisation.

Enzyme 2: AMG^®^ 300L is an amyloglucosidase that hydrolyses the (1,4) and (1,6) alpha D-glycosidic bonds at the nonreducing ends of polysaccharides, increasing the sweetness in the hydrolysed product.

#### 2.1.2. Hydrolysis Method

The enzymatic starch hydrolysis method was developed by combining information from enzyme suppliers and from the literature [27,28]. The enzymatic hydrolysis of starch was carried out in a thermostated reactor. Oat flour (Celnat, Saint-Germain Laprade, France) with a starch content of 65 g/100 g was mixed with water (1:3/flour:water) and heated to 60 °C. The first step of the hydrolysis was to decrease the viscosity of the flour/water mixture after addition of enzyme 1 (0.4 g/100 g), under stirring at a temperature of 60 °C for 15 min. The second step was the saccharification of the mixture with the addition of enzyme 2 (0.3 g/100 g), maintaining the temperature at 60 °C for 45 min. 

During the reaction, samples were taken at times 0, 2.5, 5, 10, 15, 20, 30, 40, 50, and 60 min; then, enzymes present in the resulting hydrolysate were inactivated by heat treatment at 90 °C during 5 min, before being frozen in liquid nitrogen and stored at −80 °C for further experiments.

### 2.2. Hydrolysate Characterisation

#### 2.2.1. Determination of Starch and Sugar Content

The available starch and sugar content in the hydrolysate samples was determined using an enzyme kit Total Starch HK and K-SUFRG (Megazyme, Lybios, Pontcharra-Sur-Turdine, France). Analyses were carried out on the initial flour/water mixture (at 0 min) and during the 60 min of hydrolysis.

#### 2.2.2. Conductivity Measurement

The conductivity of the hydrolysis products was analysed using a conductivity meter (Metrohm, Herisau, Suisse). The calibrated and cleaned probe was immersed in the various samples taken during hydrolysis. The conductivity was expressed in µS/cm.

### 2.3. Preparation of Fermented Products 

Oat (*Avena sativa* L.) (Celnat, Saint-Germain Laprade, France), chickpea (*Cicer arietinum*) (Priméal, Peaugres, France), and coconut flours (*Cocos nucifera*) (Ecoidées, Soultz Sous Forets, France) were selected, and three different fermented products were prepared. A concentration of flour and hydrolysate was rehydrated with tap water at 50 °C to obtain a solution of 15% dry matter (Table 1). 

The formulation and manufacturing process were selected after optimisation based on the results of a previous study [29]. Samples were pasteurised by heating at 90 °C for 10 min in the Minilab^®^ mixer. (Olsa S.p.A, Milan, Italy). This was followed by rapid cooling to 45 °C. Before inoculation, lactic acid bacteria (LAB) (0.2 DCU/kg) were resuspended for 30 min in sterile vials containing 10 mL plant juice. The flasks were shaken regularly for rehydration of the LAB strains and for obtaining a homogeneous solution without residuals. The fermentation process was stopped by cooling the product to 15 °C after 6 h. Subsequently, it was transferred to sterile cups and stored at 4 °C for 23 days. The vials were shaken regularly and gently in order to rehydrate the strains of LAB and thus to obtain a homogeneous solution free of residues. After 6 h, fermentation was stopped by cooling the product to 15 °C, which was then transferred to sterile cups and stored at 4 °C for 23 days.

### 2.4. Characterization of the Fermented Products

#### 2.4.1. Kinetics of Acidification, Determination of pH, and Titratable Acidity

Acidification kinetics can be used to monitor changes in the pH of a product. The pH tends to decrease over the incubation time of the bacterial strains until a target value is reached. pH probes coupled with temperature sensors were immersed in samples that had been placed in sterile tubes. Then, the tubes were placed in a water bath set at 45 °C, the maturation temperature of the lactic acid bacteria. The pH probes were connected to a Consort D291 module (Consort bvba, Turnhout, Belgium). The programme recorded the variation in pH with an interval of 30 min between two measurements over a total period of 6 h.

After processing and storing the fermented beverages at 4 °C, pH and acidity measurements were performed. The pH was measured at room temperature using a pH meter (Model: Mettler Toledo F20-Kit FiveEasy™ Benchtop, N.V. Mettler-Toledo S.A., Zaventem, Belgium). The determination of acidity was carried out by the potentiometric method (Afnor V04-369). A sample of approximately 10.0 ± 0.1 g was mixed with 10 mL of distilled water. The mixture was titrated with a sodium hydroxide (NaOH) solution (0.1 N), following the evolution of the pH to 8.30. The acidity was expressed as the amount of NaOH used (mL NaOH/10 g) and recorded as the titratable acidity [30].

#### 2.4.2. Texture Analysis

The texture of the fermented desserts was analysed using a Lloyd traction/compression analyser (AMETEK S.A.S, Elancourt, France). A TPA texture profile analysis procedure was conducted using a 10 N cell force. A flat base cylindrical probe (1 cm in diameter, 10 cm long) was used to compress each sample twice, down to 40% of their original height, with a crosshead speed of 2 mm/s., according to Wang and Guo’s [31] method with some modifications.

#### 2.4.3. Water-Holding Capacity

Water-holding capacity (WHC) refers to the amount of water that the sample can retain, either fully or partially. This quantification was determined through the centrifugation method, as described by Wang et al. [32], with minor adjustments. Fermented products (5 g) at 4 °C were centrifuged at 4500 rpm (5732× *g*) for 20 min. Water-holding capacity (WHC) was determined as the ratio of water retained per unit of fermented product (g/100 g), as per Equation (1).
(1)$$WHC\;\left(g/100g\right)=\frac{{Mass}_{sediment}}{{Mass}_{gel}}\times 100$$

#### 2.4.4. Microbiological Analysis

A volume of 1 mL was extracted from the fermented mixture, and then it was thoroughly mixed with 9 mL of sterilised tryptone salt broth (NaCl 0.85% (*w*/*v*)). Subsequently, a range of suitable dilutions of the sample were separately applied in triplicate on Man, Rogosa, and Sharp (MRS) and M-17 selective agar media. *Lactobacillus delbrueckii* subsp. *bulgaricus* was enumerated on MRS agar at a pH of 5.4 under controlled anaerobic conditions, incubating at 37 °C for 48 h. *Streptococcus thermophilus* was enumerated on M-17 agar incubated at 37 °C for 72 h [33]. Post-incubation, Petri dishes containing from 30 to 300 colonies were systematically counted and expressed as log_10_ colony-forming units per millilitre (log_10_ CFU/mL) of the sample.

#### 2.4.5. Quantification of Exopolysaccharides

Exopolysaccharides (EPSs) were extracted from the fermented products following the approach of Hickisch et al. [34] with slight modifications.

A sample was diluted with distilled water in a 1:1 ratio, and then 10 mL of a 20% TCA solution in 100 mL of the solution was added to precipitate the proteins. The mixture was stirred, followed by separation through centrifugation (3330× *g* at 4 °C for 30 min). The pH of the supernatant was adjusted to 6.8 using a 40% NaOH (*w*/*v*) solution. A second precipitation step was performed, followed by heating for 30 min at 100 °C. The EPS was precipitated by adding two volumes of absolute ethanol at 4 °C to the entire volume of the obtained supernatant. The EPS was collected through centrifugation (3330× *g* at 4 °C for 30 min) after overnight storage at 4 °C. The pellet was dissolved in distilled water and dialysed in tubes (Visking Dialysis Tubing from Medicell Membranes Ltd., London, UK) graded at 3500 Da (size no. 4, 35 mm, 55 mm, 5 m) in distilled water at 4 °C, and refreshed three times a day for 24 h. After two days of dialysis, the resulting solution was lyophilized and weighed.

### 2.5. Statistical Analysis

All analyses were performed in triplicate to calculate the mean and standard deviation of a random sample.

The experimental data were analysed statistically using linear regression and analysis of variance (ANOVA), with a *p*-value threshold of 0.05.

We used a post hoc Tukey pairwise comparison test to determine if there were statistically significant differences (*p* < 0.05) between the mean values of the different samples at the 95% confidence level. The results are presented as the mean ± standard deviation, and statistically significant differences are indicated by superscript letters.

## 3. Results and Discussion

### 3.1. Hydrolysate Characteristics

#### 3.1.1. Composition of the Hydrolysate

The enzymatic hydrolysis experiments were carried out on an oat flour/water mixture in a reactor with a starch content of 14 g/100 g calculated at time 0 (beginning of the reaction). This experiment showed the impact of enzymes on the degradation of starch molecules, which led to a decrease in viscosity but also to an increase in the sugar content (Figure 1).

After only 2 min of reaction, the starch content in the mixture decreases from 14.45 ± 0.11 g/100 g to 6.51 ± 0.40 g/100 g. According to Zhu [5], oat starch is highly susceptible to α-amylase digestion. This susceptibility is mainly because of the larger number of short amylopectin unit chains, leading to structural defects in granule crystallites. Furthermore, the relatively small granule size of oat starch facilitates interactions between starch and enzymes. The starch value continues to decrease slowly during the 60 min reaction time with the use of amyloglucosidase, to reach a value of 2.96 ± 0.34 g/100 g. The hydrolysis is considered to be partial as a positive concentration of starch is detected at the end of the reaction. The starch degradation was followed by a considerable increase in the sugar content from 0.12 ± 0.04 g/100 g at the beginning of the reaction to 7.87 ± 0.88 g/100 g after 60 min. Several factors affect the degree of hydrolysis and amount of sugar produced, including starch type, amylase concentration, granule size, temperature, pH, and substrate/enzyme ratio [21,35,36]. A study by Abebe et al. [37] observed a significant increase in the rapidly available glucose content and decrease in starch fractions in teff flour, which is directly related to flour size. According to the authors, the starch digestibility of ground cereal flours increases with decreasing flour size. 

The statistical analysis showed a highly significant evolution of the sugar (*p* < 2 × 10^−16^) and starch (*p* < 1.339 × 10^−5^) curves in function of the reaction time. Each curve profile is characterised by an initial high reaction rate, followed by a rapid decrease in rate which tends to stabilise (Figure 1). The evolution of the curves can be attributed to different mechanisms, considering a progressive decrease in the number of starch bonds that can be hydrolysed, the inhibition of the enzyme by the reaction products, or the denaturation of the amylase by the action of temperature.

#### 3.1.2. Conductivity Variation 

Conductivity is the ability of the hydrolysate solution to carry an electric current. In a solution, it is the anion and cation balance that carries the current.

The electrical conductivities of the hydrolysate shown in Figure 2 decrease at different times of hydrolysis. Conductivity values change from 295.10 ± 8.48 µS/cm at 2 min of hydrolysis to 281.53 ± 1.07 µS/cm at 60 min of hydrolysis.

The decrease in charge density is very significant (*p* < 0.001) over time. This decrease may have been caused by the large number of starch molecules present in the solution, degraded by enzymatic hydrolysis. In a study of enzymatic hydrolysis of maize starch using different methods, the authors found no significant difference in conductivity (103–119 µS/cm) during the first 24 h [38]. According to Soria-Hernández et al. [39], foods containing electrolytes, such as salts, acids, and certain gums and thickeners, contain charged groups and have a significant effect on conductivity. Conductivity is important in food and beverage development because of its influence on physical parameters, the interactions with other ingredients, and the stability of proteins in a given food system.

### 3.2. Characterisation of the Fermented Product

#### 3.2.1. Kinetics of Acidification

The acidification kinetics are presented in Figure 3, which shows the variation in pH during fermentation of the P1 mixture compared to the P2 and P3 mixtures.

In the three products, the pH values are quite close. A decrease in pH and an increase in the number of lactic acid bacteria are promising indicators of fermentation [14]. The lactic acid bacteria reduced the pH from 6.28 to 6.44 to reach values between 4.69 and 4.85 after 6 h of fermentation. In a study by Masiá et al. [40], the authors indicate that the time required to reach a pH of 4.5 depends on the raw material used and the glucose content, i.e., (5.1 to 5.9 h) for a coconut-based product, (6.6 to 8.5 h) for an oat-based product, and (6.0 to 7.0 h) for a soya-based product. This corresponds to the results obtained with a lower pH at the end of fermentation in P3, which is characterised by a higher sugar content provided by the coconut flour and the oat hydrolysate. 

#### 3.2.2. Post-Acidification of Fermented Products

Post-acidification was determined by measuring pH and acidity at 1, 7, 15, and 23 days after conditioning as shown in Table 2. 

The pH of yoghurt stored at 4 °C is below 4.5, which causes post-acidification, due to the presence of lactic acid bacteria that reduce the pH from 0.5 to 1.0, and the level of acidity increases slightly from 0.2% on the first day to 0.5% on the 28th day of storage [14]. The obtained results show the presence of post-acidification, with a highly significant effect (*p* < 0.05) of hydrolysate addition and storage time in the three products. pH and acidity varied from 4.40 ± 0.02 to 4.47 ± 0.02 pH unit and 4.73 ± 0.06 to 5.03 ± 0.06 mL NaOH/10 g at day 1 to the range of 4.27 ± 0.01 to 4.41 ± 0.01 pH unit and 5.53 ± 0.06 to 6.70 ± 0.10 mL NaOH/10 g after 23 days of storage at 4 °C, respectively. The results were consistent with those of Gupta et al. [41] who observed a decrease in pH (4.00) and an increase in acidity (12%) during 3 weeks (21 days) of storage on oat products. The same results were obtained in a study by Duru et al. [30], which suggests that acid production by lactic acid bacteria during 28 days of storage resulted in a decrease in the pH and an increase in titratable acidity of fermented oat products. According to Gallo et al. [42], as the concentration of glucose increases, the production of lactic acid increases significantly in fermented plant-based products. This may explain the significant post-acidification in hydrolysate products due to the higher sugar concentration.

#### 3.2.3. Texture Analysis

Texture analysis allows to translate the thickness of the gel in the three different products after fermentation. In the present study, the analysis also showed the strength and impact of the breakage applied to the gels. The texture of the products during the storage period is characterised by three distinct parameters: hardness, elasticity, and cohesion (Table 2).

The hardness provided represents the force required to break the gel. The statistical analysis revealed a significant effect of product type and storage on hardness. The hardness values were the most important in product P1 (0.19 ± 0.01 N) and were equal in products P2 and P3. These measurements increase in all products during the storage period to reach a maximum value of 0.55 ± 0.04 N. The same evolution of firmness was recorded in a fermented dessert made from whey and ingredients from the bark of the Jabuticaba tree in a study by Almeida Neta et al. [43]. The authors report a significant (*p* < 0.05) increase in firmness from a value of 0.60 ± 0.24 to 0.89 ± 0.38 N on the first day of storage to a value of 0.81 ± 0.27 to 0.96 ± 0.38 N after 21 days of storage. According to Genevois et al. [44] and Demir et al. [6], firmness is directly related to the total amount of solids and protein content in fermented milk products, such as yoghurt, resulting in a gel with a dense and rigid structure. However, the textural characteristics of plant-based products are influenced more by the use of hydrocolloids than by the low protein content and by the interaction between starch and hydrocolloids [1]. This may explain the low hardness in the products with hydrolysate (P2 and P3), due to the lower solids and the particularly low starch content in these products.

Cohesion is an index that corresponds to the capacity of the sample to resist two successive compressions. The statistical analysis showed no significant effect of the hydrolysate or the storage time on cohesion. Contrary to the hardness measurement, the cohesion decreases with the storage period with a maximum value that can reach 0.53 ± 0.08 at day 1 and a minimum value of 0.24 ± 0.02 after 23 days of storage. The cohesion of the three products showed almost the same cohesion as yoghurts made from dairy products and soya [45]. However, these values are lower than those reported by Raikos et al. [46] for a product based on oat supplemented with aquafaba and vegetable oil, which were as high as 0.69. This force was necessary to rupture the gel, which had greater density and resistance due to the interaction between the fat globules and protein matrix. In this study, the higher cohesion in P1 can be explained by the presence of a high level of starch in this product compared to P2 and P3, which may contribute to a firmer gel.

Elasticity is an index that defines the height at which the sample returns to its original shape after the first compression in relation to the maximum deformation. The values of elasticity ranged from 11.10 ± 1.27 mm to 18.03 ± 0.70 mm. The values were quite similar to a product made from oat [46]. The statistical analysis revealed a significant effect (*p* < 0.05) of product type and storage on elasticity. In light of this, texture variation can be due to the presence and retrogradation of starch. According to a study by Kumar et al. [35] on a milk–barnyard millet product, it was indicated that millet starch tends to retrograde during cooling, which gives the product a pasty texture typical of native starches, as opposed to an enzymatically hydrolysed starch. The authors suggest that the equal mixture of milk and hydrolysate improves the texture of the product in a phase separation due to milk–polysaccharide interactions.

#### 3.2.4. Water-Holding Capacity

This analysis determined the capacity of the gel to retain water incorporated into the mixtures during production over 23 days of storage (Table 2). Syneresis represents the expulsion of water from a gel. This film of water on the surface affects the acceptability of the probiotic product by consumers. Water-holding capacity (WHC) is a useful way to describe the ability of a food matrix to retain free water when an external force is applied [44]. The addition of hydrolysate and the storage period both have a highly significant effect (*p* < 0.05) on the evaluation of the final WHC content. After preparation (day 1), water retention is 100% in product P1, which contains only flour-type ingredients. However, the addition of hydrolysate to the P2 and P3 mixes decreases this water retention to 72.22 ± 0.50 and 68.67 ± 1.25%, respectively. These values tend to decrease during the 23 days of storage at 4 °C, about 10% in product P3 and 12% in product P2, and up to 27% in product P1. According to Duru et al. [30], a low WHC in an oat flour product may be related to a lower viscosity. The analysis of these values shows an increase in water retention in the product with a higher flour content and decreases with an increasing hydrolysate content. As the starch granules increase, their viscosity increases, allowing them to retain more water and reduce syneresis [45]. According to Demїr et al. [6], there is a link between improved texture of dairy yoghurt and increased total solid content on water-holding capacity. These authors also indicate that starch has a major role in improving the water-holding capacity and reducing syneresis. Gularte and Rosell [47] also indicated positive correlations between starch digestibility and physicochemical properties, such as final viscosity and WHC reduction.

#### 3.2.5. Microbiological Analysis

The enumeration of lactic acid bacteria strains was carried out during the 23 days of storage (Figure 4).

During the first day of storage, the number of *Streptococcus thermophilus* bacteria in all products was approximately equal to values ranging from 8.61 ± 0.03 to 8.65 ± 0.02 log_10_ CFU/mL and more divergent for *Lactobacillus delbrueckii* subsp. *bulgaricus* ranging from 5.36 ± 012 to 6.15 ± 0.01 log_10_ CFU/mL. According to Gallo et al. [42], glucose concentrations during fermentation time have a direct impact on the final bacterial load of an oat flour product. Thus, the authors observed that the microbial load was always higher after the addition of glucose than in the absence of glucose. But, glucose concentrations above 5% led to substrate inhibition, at least with respect to bacterial growth. As a result, the concentration of glucose in the different products was sufficient for the development of the LAB and the reduction in the pH during the 6 h of fermentation. However, this concentration could be not sufficient (in the hydrolysis products) for the simultaneous development of the two LAB stains, which can explain a lower number of *Lactobacillus delbrueckii* subsp. *bulgaricus*. A decrease was observed during the 23 days of enumeration of both strains. According to Demir et al. [6], yoghurt cultures should maintain viability above 6–7 log_10_ CFU/g until the end of shelf life during cold storage. The loss of viability of LAB in oat milk yoghurt is the result of acid damage to these microorganisms. The variation in the *Streptococcus thermophilus* population in time was in agreement with that recorded in the literature [43,48], which indicates a significant reduction around 0.10 log_10_ CFU/g after day 21. The survival of *Lactobacillus* spp. is usually reduced to lower counts by the 14th day of storage [8].

During storage, the number of *Streptococcus thermophilus* and *Lactobacillus delbrueckii* subsp. *bulgaricus* could be affected by the chemical composition of the ingredients and a decrease in pH level. The high acid production and reduced pH observed in the fermented samples after four weeks of storage could explain the loss of viability of the probiotic strains [30,49]. This is in line with the acidity and pH results obtained in Table 2. The acidity recorded during the storage period is higher in P3, followed by P2 and P1. This may explain a decrease in the bacterial count (*Lactobacillus delbrueckii* subsp. *bulgaricus*), despite the addition of hydrolysate. These results are supported by the study by Jaster et al. [50], which indicates a significant reduction (*p* < 0.05) in the LAB count during the storage period in yoghurt enriched with strawberry pulp concentrate, due to the decrease in pH and the increase in acidity. The authors specified that some starter microorganisms may disappear, in a pH below 4.2, contributing to the decrease in the LAB count. Nevertheless, even with some slight variations in the LAB count in each storage period studied (less than one cycle log), the recommended LAB count was maintained. 

Szydłowska et al. [51] report a number of lactic acid bacteria below 8.34–8.40 log_10_ CFU/g after 21 days of storage in cereal-based desserts, which can therefore be considered as potentially probiotic and/or symbiotic products. However, the value of 10 million bacteria per gram relating to the milk part is also reported by Article 2 of “Décret n°88-1203 du 30 décembre 1988” on fermented milk and yoghurt in French legislation. 

The statistical analysis indicates a highly significant (*p* < 0.05) effect of hydrolysate concentration on *Streptococcus thermophilus* (more important in product P3) and *Lactobacillus delbrueckii* subsp. *bulgaricus* viability (more important in product P1). This difference is probably due to the sugar composition of the products. The interactions between the two strains are described as proto-cooperation. *Streptococcus thermophilus* produces CO_2_ and formic acid that stimulate the growth of *Lactobacillus delbrueckii* subsp. *Bulgaricus,* which hydrolyses milk proteins releasing peptides and amino acids to enhance the growth of *Streptococcus thermophilus* [52,53].

#### 3.2.6. Exopolysaccharide Performance

Exopolysaccharides (EPSs) are biopolymers, which are synthesised and excreted by the microbial strains used in the fermentation process. An assay of EPS was carried out at different storage periods (from 1 to 23 days) of the P1, P2, and P3 products. The analysis aimed to verify the capacity of the two strains contained in the ferment (*Lactobacillus delbrueckii* subsp. *bulgaricus* and *Streptococcus thermophilus*) to produce EPS in a plant-based substrate and to compare the concentrations of EPS between the presence and absence of hydrolysis in the formulation. The results of this analysis are shown in Figure 5.

Firstly, the experiment confirmed the ability of the lactic acid bacteria strains used to produce exopolysaccharides. The production of EPS by lactic acid bacteria in yoghurts allows the improvement of the texture of the products [30]. Secondly, the concentrations of EPSs vary from one product to another due to differences in their composition. After day 1 of storage, the EPS values were higher in P1 (0.73 ± 0.04 g/100 g), twice as much as P2 (0.37 ± 0.11 g/100 g), and three times as much as P3 (0.23 ± 0.05 g/100 g). The two lactic acid bacteria allowed high levels of EPS in all products compared to those reported in the literature. Previous studies had indicated EPS levels between 0.03 and 0.66 g/100 g produced by different lactic acid bacteria in formulations based on selected cereals and legumes, such as quinoa, lupin, and chickpea [34,54,55]. The concentration of EPS generally varies according to the extraction method. According to Comte et al. [56], chemical methods extract 9 to 12 times more EPS than physical methods. The statistical analysis showed a highly significant effect (*p* < 0.05) of product type and the hydrolysate addition on EPS production in the three products. However, EPS production for the same product over time of storage at 4 °C was only significant in P3 and P1 when comparing days 1 and 23. Luana et al. [57] and Peyer et al. [22] have stated that the type and amount of EPS produced depends mainly on the sugars present in the medium, but also on the presence of micronutrients and environmental conditions (temperature and incubation time). As glucose was a predominant sugar in hydrolysates, it can be concluded that this type of sugar is less effective in producing EPS during fermentation, but this stabilises the concentration during storage. The storage period had a nonsignificant effect on the concentration of EPS, despite a decrease in product P1 and an increase in products P2 and P3, reaching 0.45 ± 0.08, 0.43 ± 0.20, and 0.35 ± 0.08 g/100 g, respectively, at 23 days. The production of EPS in the P1 product follows the same trend as indicated in a study by Li et al. [58] with a decrease in the quantities of EPS over the 21 days of storage at 4 °C. This can result from the reduced concentration of nutrients available for the growth of lactic acid bacteria. Ramchandran and Shah [59] made similar observations to the P2 and P3 products, with a decrease in EPS content on day 7 of storage, explained by the presence of enzymes capable of degrading the EPS, followed by a slight increase over the following 3 weeks. Zhao and Liang [60] also found an increase in EPS in yoghurt samples after 21 days of storage. The authors attributed this increase to synergistic and antagonistic interactions between the lactic acid bacteria. Variations in the EPS estimation method, differences in EPS types, and variations in deformation could be the possible reasons for the observed variations [59].

## 4. Conclusions

This study investigated the influence of partial oat flour hydrolysis on quality, flavour, physicochemical properties, and microbiological impact of a fermented dessert based on a mix of oat, chickpea, and coconut flour. Thus, it shows the potential of incorporating oat flour hydrolysate into a fermented vegetal dessert in order to improve sweetness by increasing sugar without additives while maintaining good texture properties and exopolysaccharide production even if they are lower than the formulation without hydrolysate. These results provide a basis for testing new ingredients in the formulation of new plant-based desserts. By combining the information obtained, it is possible to work on the hydrolysates to provide better solutions for formulating these desserts without adding additives.

## Figures and Tables

**Figure 1 foods-12-03868-f001:**
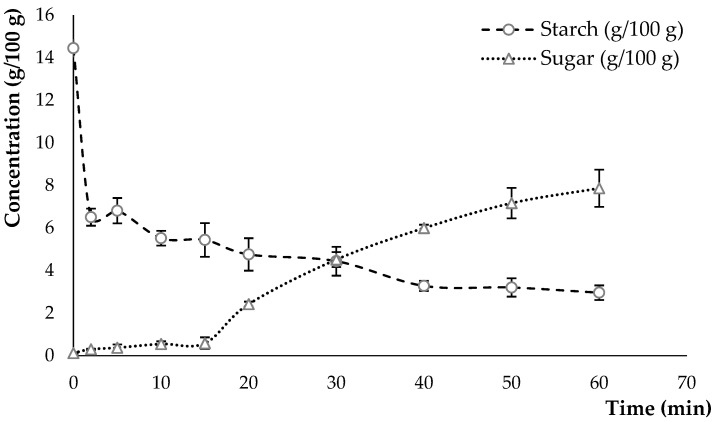
Starch degradation and sugar concentration in an oat flour/water solution during enzymatic hydrolysis at 60 °C for 60 min.

**Figure 2 foods-12-03868-f002:**
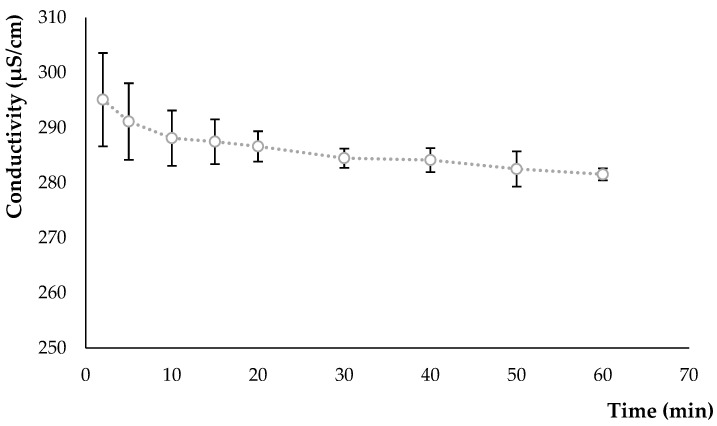
Evolution of conductivity in an oat flour/water solution during enzymatic hydrolysis at 60 °C for 60 min.

**Figure 3 foods-12-03868-f003:**
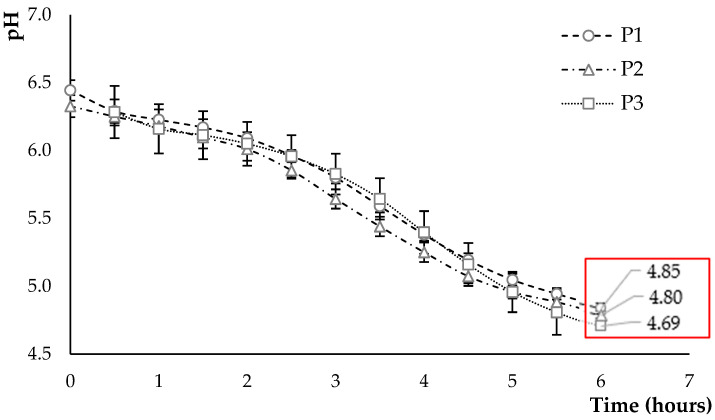
Acidification kinetics during 6 h of fermentation with lactic acid bacteria strains. P1: 100% oat flour and 0% oat hydrolysate; P2: 50% oat flour and 50% hydrolysate; P3: 30% oat flour and 70% hydrolysate. Red box gives mean pH-value obtained for each formulation after 6 hours of fermentation.

**Figure 4 foods-12-03868-f004:**
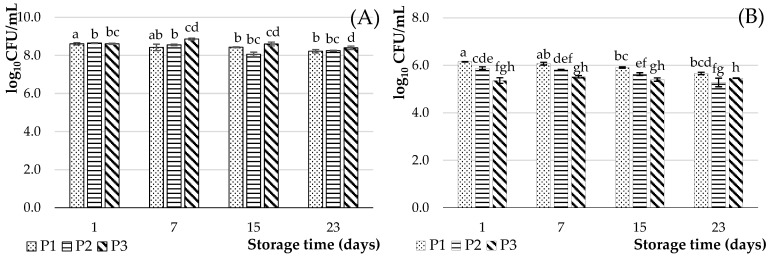
Cell counts (log_10_ CFU/mL) of lactic acid bacteria (**A**): *Streptococcus thermophilus* and (**B**): *Lactobacillus delbrueckii* subsp. *bulgaricus* in plant-based products fermented for 6 h at 45 °C and stored for 23 days at 4 °C. P1: 100% oat flour and 0% oat hydrolysate; P2: 50% oat flour and 50% oat hydrolysate; P3: 30% oat flour and 70% oat hydrolysate. Values of different groups of letters are significantly different (*p* < 0.05).

**Figure 5 foods-12-03868-f005:**
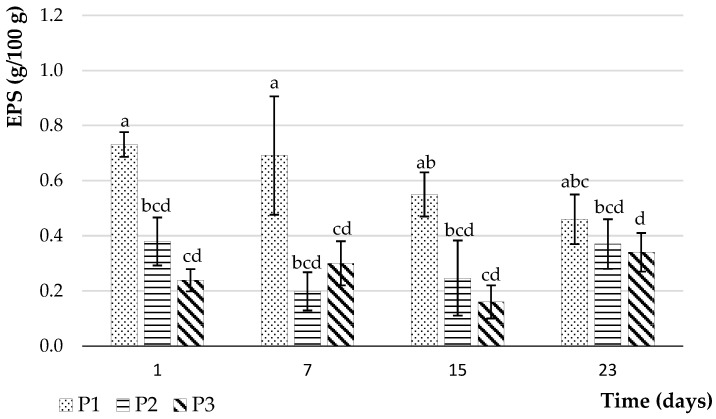
Evaluation of the exopolysaccharide (EPS) yields in P1, P2, and P3 at different storage times (1, 7, 15, and 23 days) at 4 °C. P1: 100% oat flour and 0% oat hydrolysate; P2: 50% oat flour and 50% oat hydrolysate; P3: 30% oat flour and 70% oat hydrolysate. Values of different groups of letters are significantly different (*p* < 0.05).

**Table 1 foods-12-03868-t001:** Dry matter content of the ingredients in the three fermented products developed during the study.

Ingredients	Product 1	Product 2	Product 3
Oat flour	3.0%	1.5%	0.9%
Oat hydrolysate	-	1.5%	2.1%
Chickpea flour	9.0%	9.0%	9.0%
Coconut flour	3.0%	3.0%	3.0%

**Table 2 foods-12-03868-t002:** Characterisation of post-acidification, texture, and water retention of fermented products at 1, 7, 15, and 23 days of storage at 4 °C.

	Storage Time (Days)	pH	Acidity	WHC (%)	Hardness (N)	Cohesion	Elasticity (mm)
P1	1	4.45 ± 0.01 ^a^	4.77 ± 0.15 ^ab^	100.00 ± 0.00 ^a^	0.19 ± 0.004 ^a^	0.50 ± 0.02 ^a^	17.58 ± 0.53 ^a^
	7	4.36 ± 0.01 ^ab^	4.47 ± 0.06 ^ab^	83.07 ± 1.29 ^b^	0.26 ± 0.01 ^b^	0.32 ± 0.10 ^ab^	14.27 ± 0.84 ^ab^
	15	4.43 ± 0.03 ^bc^	5.37 ± 0.21 ^cd^	80.13 ± 0.11 ^b^	0.35 ± 0.02 ^b^	0.33 ± 0.05 ^abc^	15.89 ± 0.39 ^abc^
	23	4.41 ± 0.01 ^cd^	5.53 ± 0.06 ^c^	72.57 ± 0.21 ^c^	0.55 ± 0.04 ^b^	0.36 ± 0.02 ^abcd^	18.03 ± 0.70 ^abcd^
P2	1	4.40 ± 0.02 ^cde^	5.03 ± 0.06 ^ad^	72.22 ± 0.50 ^c^	0.09 ± 0.004 ^c^	0.46 ± 0.07 ^bcd^	13.19 ± 0.48 ^abcde^
	7	4.28 ± 0.02 ^cde^	4.60 ± 0.10 ^ad^	65.99 ± 0.93 ^d^	0.18 ± 0.01 ^cd^	0.33 ± 0.06 ^bcd^	13.90 ± 1.02 ^bcdef^
	15	4.38 ± 0.01 ^de^	6.00 ± 0.10 ^e^	68.44 ± 2.43 ^d^	0.23 ± 0.02 ^cd^	0.27 ± 0.04 ^cd^	12.52 ± 1.07 ^cdef^
	23	4.27 ± 0.01 ^e^	6.37 ± 0.15 ^b^	63.28 ± 1.23 ^de^	0.31 ± 0.01 ^cd^	0.26 ± 0.10 ^d^	11.21 ± 0.63 ^cdef^
P3	1	4.47 ± 0.02 ^f^	4.73 ± 0.06 ^ef^	68.67 ± 1.25 ^def^	0.09 ± 0.002 ^d^	0.53 ± 0.08 ^d^	16.82 ± 1.48 ^def^
	7	4.39 ± 0.01 ^fg^	5.03 ± 0.12 ^b^	65.26 ± 0.81 ^ef^	0.19 ± 0.004 ^d^	0.26 ± 0.01 ^d^	13.54 ± 0.72 ^ef^
	15	4.30 ± 0.01 ^fg^	6.30 ± 0.17 ^fg^	68.27 ± 0.38 ^fg^	0.20 ± 0.01 ^e^	0.23 ± 0.01 ^d^	11.10 ± 1.27 ^f^
	23	4.25 ± 0.01 ^g^	6.70 ± 0.10 ^g^	62.12 ± 0.49 ^g^	0.33 ± 0.005 ^e^	0.24 ± 0.02 ^d^	15.42 ± 2.61 ^f^

All analyses were performed in triplicate. P1: 100% oat flour and 0% oat hydrolysate; P2: 50% oat flour and 50% hydrolysate; P3: 30% oat flour and 70% hydrolysate. Values in the same column with different groups of letters are significantly different (*p* < 0.05).

## Data Availability

The data presented in this study are available on request from the corresponding author.

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
