# Peer review of "Impact of a Starch Hydrolysate on the Production of Exopolysaccharides in a Fermented Plant-Based Dessert Formulation"

_foods, 2023, doi:10.3390/foods12203868_

Round 1
Reviewer 1 Report
Comments and Suggestions for Authors
I reviewed the manuscript titled “Impact of a starch hydrolysate on the production of exopolysaccharides in a fermented plant-based dessert formulation. The content is novel and meets the journal criteria. However, authors should consider suggestions below to improve the quality of the manuscript.
Remove the full stop after a title
Abstract
Background of the study must be introduced
Objectives’ should be clearly identified
Conclusions should be included in abstract
Introduction
Research objectives must be past tense
Why have authors selected those specific enzymes?
2.1.2. Hydrolysis method: provide citation for this method
Preparation of fermented products: provide citation for this method
2.4. Characterization of the fermented product: provide citation for this method
Many methodology sections are without relevant citation
2.4.2. Texture analysis: provide citation
2.4.4. Microbiological analysis: provide citation
Quality of figures must be improved throughout the manuscript
Line 213: goes from….. please revise the sentence
Figure 3. please perform the statistical analysis for better understanding
Lines 256-257: which is (from 5.1 to 5.9 h) in coconut, (6.6 to 8.5 h) for oat and (6.0 to 7.0 h) for soybean. Please revise the sentence. As such, it is difficult to understand. It could be presented n a better way
Table 1: statistical notations must be defined at the table footnote
Texture analysis: discussion must be improved in this. Please compare with available literature and provide reasons.
3.2.4. Water holding capacity
Figure 5: X and Y axis should be shown clearly. 7,0 and 7,5… etc should be changed to English notations as 7 and 7.50 etc. Please revise this throughout the manuscript.
Figure 6: X and Y axis should be shown clearly. 0,0 and 0,6… etc should be changed to English notations as 0and 0.60 etc. Please revise this throughout the manuscript. Statistical analysis should be performed and denoted at the fig footnote
Conclusions
Authors divided many small paragraphs. I suggest to keep reasonable paragraphs instead of very small paragraphs
References must be in line with J format
Comments on the Quality of English LanguageI reviewed the manuscript titled “Impact of a starch hydrolysate on the production of exopolysaccharides in a fermented plant-based dessert formulation. The content is novel and meets the journal criteria. However, authors should consider suggestions below to improve the quality of the manuscript.
Remove the full stop after a title
Abstract
Background of the study must be introduced
Objectives’ should be clearly identified
Conclusions should be included in abstract
Introduction
Research objectives must be past tense
Why have authors selected those specific enzymes?
2.1.2. Hydrolysis method: provide citation for this method
Preparation of fermented products: provide citation for this method
2.4. Characterization of the fermented product: provide citation for this method
Many methodology sections are without relevant citation
2.4.2. Texture analysis: provide citation
2.4.4. Microbiological analysis: provide citation
Quality of figures must be improved throughout the manuscript
Line 213: goes from….. please revise the sentence
Figure 3. please perform the statistical analysis for better understanding
Lines 256-257: which is (from 5.1 to 5.9 h) in coconut, (6.6 to 8.5 h) for oat and (6.0 to 7.0 h) for soybean. Please revise the sentence. As such, it is difficult to understand. It could be presented n a better way
Table 1: statistical notations must be defined at the table footnote
Texture analysis: discussion must be improved in this. Please compare with available literature and provide reasons.
3.2.4. Water holding capacity
Figure 5: X and Y axis should be shown clearly. 7,0 and 7,5… etc should be changed to English notations as 7 and 7.50 etc. Please revise this throughout the manuscript.
Figure 6: X and Y axis should be shown clearly. 0,0 and 0,6… etc should be changed to English notations as 0and 0.60 etc. Please revise this throughout the manuscript. Statistical analysis should be performed and denoted at the fig footnote
Conclusions
Authors divided many small paragraphs. I suggest to keep reasonable paragraphs instead of very small paragraphs
References must be in line with J format
Author Response
Please see the attachement

Reviewer 2 Report
Comments and Suggestions for Authors
The research has studied the impact of adding hyrolysates on quality parameters of fermented desert made from composite flour. The information presented will be valuable for new product development.
There introduction and result and discussion sections will benefit greatly if re-worded to be more precise and focussed. For instance in the introduction, what is the reason for the use of mixed flour, what inform the reason for making Oat hydrolysate but not others.
There is room for improvement.
There are some typos in sub-headings.
There are number of places where terminology were used before being defined.
Reviewer 3 Report
Comments and Suggestions for Authors
Dear Editors and authors,
1-The abstract of the manuscript needs to be supported by some important results that have been obtained.
2- Most of the work methods do not contain scientific references. I was not able to access the method and make sure of it. I suggest some references that correspond to the methods used
(Jridi, M., Souissi, N., Salem, M. B., Ayadi, M. A., Nasri, M., & Azabou, S. (2015). Tunisian date (Phoenix dactylifera L.) by-products: Characterization and potential effects on sensory, textural and antioxidant properties of dairy desserts. Food Chemistry, 188, 8-15.) for analysis of the texture of the fermented desserts
(Al-Sahlany, S. T. G., Khassaf, W. H., Niamah, A. K., & Abd Al-Manhel, A. J. (2023). Date juice addition to bio-yogurt: The effects on physicochemical and microbiological properties during storage, as well as blood parameters in vivo. Journal of the Saudi Society of Agricultural Sciences, 22(2), 71-77.) for Microbiological analysis.
(Evdokimov, I. A., Volodin, D. N., Misyura, V. A., Zolotorevа, М. S., & Shramko, М. I. (2015). Functional fermented milk desserts based on acid whey. Foods and Raw Materials, 3(2), 40-48.) analysis of Water holding capacity the fermented desserts.
3-Figures 2, 3, 4, and 6 are devoid of statistical analysis and some small letters must be added to differentiate between coefficients.
4-The figures in the manuscript are not arranged and the titles of the axes are irregular. More attention should be paid to these figures because they represent the results of the manuscript.
5-The conclusions are long and contain many results.
6-The writing of the bacteria Lactobacillus delbrueckii subsp.Bulgaricus is wrong in most chapters of the manuscript. Note the summary, conclusions and methods of work. It should be written correctly and in the following form Lactobacillus delbrueckii subsp.bulgaricus.
7-The numbers written in figures 5 and 6 must be written correctly.
Comments on the Quality of English LanguageQuality of English Language is good.
Reviewer 4 Report
Comments and Suggestions for Authors
In this study, the authors did a lot of experiment for fermented plant-based dessert. Plant-based desserts are now an alternative to traditional desserts. It is a valuable study for reference. The discussion of results is insufficient and the representativeness of assessment methods is not fully discussed. These two parts should be supplemented and revised.
#1. In Line 67-73, the enzymatic hydrolysis is an important method used in this study, please add a related description in the Introduction and cite proper references.
#2. In Line 81-82, Chickpeas, oats and coconut flour were chosen as the materials for this study, but there is no explanation why these three were chosen as the main materials among the many plant-based materials. Please add the reasons for the selection and cite proper references.
#3. In Line99-105, the enzymatic hydrolysis method of starch is described. However, no references have been presented to demonstrate that the conditions of processing time and temperature used are optimal. Please add references to show why such conditions were used in this study.
#4. Figure 1 illustrates the concentration of flour and hydrolysate used in this study, same as point 3, please add a reference to explain the conditions used in this study is an optimal condition.
#5. In 2.4.2 of Materials and Methods, the conditions such as 10 N cell force and cylindrical probe used in this study. Please add references as to why these conditions were used.
#6. In 2.4.4 of Materials and Methods, the selective MRS is used for inoculated Lactobacillus delbrueckii subsp. Bulgaricus. Please state whether anaerobic culture is used in the process of lactobacilli culture, and if general aerobic culture is used, please state why anaerobic culture is not used.
#7. In Line 231-232, only the results of the cited references are stated, but not compared and discussed with the results of this study. In the subsequent discussions in this study, there were several occurrences of such a situation. Please add more comparison and discussion berween the references and the results of this study.
#8. In Line 239-241, it showed “This decrease may have been caused by the large number of starch molecules present in the solution, degraded by enzymatic hydrolysis.”. Please add references to verify this inference.
#9. In Line 253-254, it showed “The lactic acid bacteria reduced the pH from a range of 6.28 to 6.44”. However, the pH values seen in the Figure 4 are all higher than 5, is there an error in the Figure?
#10 In Line 278-279, it showed “as the concentration of glucose increases, the production of lactic acid increases significantly in a fermented plant-based products.”. However, the results of glucose analysis are not presented here. And the method of analysis of glucose is mentioned in Materials and Methods, but the results are not seen in the article.
#11 In Line 301-304, the discussion here is about cohesion. Please add relevant discussions and references cited.
#12 In Line 311-312, it showed “the equal mixture of milk and hydrolysate improves the texture of the product”. However, this phenomenon is not clearly visible from the data in the chart. In addition, the article does not state what kind of change in the texture of the product represents an improvement.
#13 In Line 331, it showed “starch has a major role in improving the water holding capacity and reducing syneresis”. However, the WHC of the products hydrolyzed from this study (P2 & P3) were deteriorated, please explain why such results were used if the products were of poor quality?
#14 It can be seen from Figure 5 that the number ofLactobacillus delbrueckii subsp. Bulgaricus is lower than the standard value. Please provide a reasonable explanation for this situation. Generally, anaerobic fermentation is used to determine the lactic acid bacteria count. Is it possible that the culture of anaerobic fermentation is not used?
#15 In Line 365-366, it showed “microbial load was always higher after the addition of glucose than in the absence of glucose”. However, the growth of the two microorganisms measured in this study showed an opposite trend to that described here. Please provide a reasonable explanation with references.
#16 In Line 394-395, it showed “but this stabilises the concentration during the storage”. However, the EPS content shown in the graph is actually decreasing before increasing. Please provide a reasonable explanation with references.
Comments on the Quality of English LanguageIt is suggested to recheck the grammar and spelling used in the manuscript.
Round 2
Reviewer 3 Report
Comments and Suggestions for Authors
Dear Editors,
After reviewing the corrected manuscript submitted by the authors, The authors did not make the required corrections. The manuscript still lacks many scientific standards.
Comments on the Quality of English LanguageQuality of English Language needs moderate corrects of English grammar.
Author Response
Please find comments on attached file

Reviewer 4 Report
Comments and Suggestions for Authors
Q14. Microbial load was always higher after the addition of glucose than in the absence of glucose”. Explanation why the growth of the two microorganisms measured in this study showed an opposite trend to that described here.
A: We observe a slightly higher number of Streptococcus thermophilus in product 2 and 3.
Q: A slightly higher number of Streptococcus thermophilus in product 2 and 3 can be observed. However, you still have not explained and discussed the phenomenon that the addition of hydrolysate to Lactobacillus delbrueckii subsp. bulgaricus resulted in a decrease in the bacterial count.
Q15. It showed “but this stabilises the concentration during the storage”. However, the EPS content shown in the graph is actually decreasing before increasing. Please provide a reasonable explanation with references.
A: Most studies on the production of EPS that we have studied have not carried out an analysis of the EPS during a period of storage but have sought to determine the yield after storage, which makes it difficult to compare our results with the literature (Line 466-476).
Q: If the relevant references does not study the EPS during the storage period, it would be appreciated if you could provide a reasonable explanation or speculation for this particular phenomenon.
Author Response
Please find comments on attached file.
